# Long-Term Results of a Standard Algorithm for Intravenous Port Implantation

**DOI:** 10.3390/jpm11050344

**Published:** 2021-04-24

**Authors:** Ching-Feng Wu, Jui-Ying Fu, Chi-Tsung Wen, Chien-Hung Chiu, Ming-Ju Hsieh, Yun-Hen Liu, Hui-Ping Liu, Ching-Yang Wu

**Affiliations:** 1Department of Surgery, Thoracic and Cardiovascular Surgery Division, Chang Gung Memorial Hospital, Linkou 333423, Taiwan; maple.bt88@gmail.com (C.-F.W.); b9102067@cgmh.org.tw (C.-H.C.); hsiehmj2@cgmh.org.tw (M.-J.H.); l571011l@cgmh.org.tw (Y.-H.L.); tpeclcra@gmail.com (H.-P.L.); 2Medical Department, Medical College, Chang Gung University, Taoyuan 333323, Taiwan; juiing0917@hotmail.com (J.-Y.F.); ctwen2001@gmail.com (C.-T.W.); 3Pulmonary and Critical Care Medicine, Department of Internal Medicine, Chang Gung Memorial Hospital, Linkou 333423, Taiwan; 4Thoracic and Cardiovascular Surgery Division, Department of Surgery, New Taipei Municipal Tu-Cheng Hospital, New Taipei City 236017, Taiwan

**Keywords:** intravenous port, totally implantable vascular access device, standard algorithm

## Abstract

Intravenous ports serve as vascular access and are indispensable in cancer treatment. Most studies are not based on a systematic and standardized approach. Hence, the aim of this study was to demonstrate long-term results of port implantation following a standard algorithm. A total of 2950 patients who underwent intravenous port implantation between March 2012 and December 2018 were included. Data of patients managed following a standard algorithm were analyzed for safety and long-term outcomes. The cephalic vein was the predominant choice of entry vessel. In female patients, wire assistance without use of puncture sheath was less likely and echo-guided puncture via internal jugular vein (IJV) with use of puncture sheath was more likely to be performed, compared to male patients (*p* < 0.0001). The procedure-related complication rate was 0.07%, and no pneumothorax, hematoma, catheter kinking, catheter fracture, or pocket erosion was reported. Catheter implantations by echo-guided puncture via IJV notably declined from 4.67% to 0.99% (*p* = 0.027). Mean operative time gradually declined from 37.88 min in 2012 to 23.20 min in 2018. The proposed standard algorithm for port implantation reduced the need for IJV echo-guided approach and eliminated procedure-related catastrophic complications. In addition, it shortened operative time and demonstrated good functional results.

## 1. Introduction

An intravenous port provides secure vascular access for delivery of treatment in cancer patients. Major issues related to intravenous port implantation include first attempt success rate [1,2,3] and peri-operative and late complications [4,5,6,7]. Several native vessels can be used as the entry vessel for chest port insertion, including the cephalic vein [2,3], deltoid branch of the thoracoacromial vein [8], the axillary vein [1,9], the internal jugular vein [3], the external jugular vein [2], the left brachiocephalic vein [10,11], and the subclavian vein [12,13,14]. However, different implantation methods are used for different target vessels and varying long-term results have been reported. There has been no consensus on recommendations because most clinical practitioners just consider intravenous ports as vascular access instead of seeing them as part of an integrated cancer treatment plan. As cancer treatment has improved, the possibility of secondary malignancy and the need for port re-implantation have gradually increased. This highlights the important role of patients’ native vessels, even with suboptimal quality, and of preserving the entry vessels for port re-implantation. Therefore, surgeons need an implantation recommendation that not only has good functional result but is also easy to follow and allows quantifiable quality control. From the literature review, vessel cutdown has been shown to have a low immediate complication rate [15] because the patient’s native vessel is explored and can be directly visualized for catheter implantation. The cephalic vein is the target vessel for vessel cutdown because of its superficial location. However, alternative entry vessels may be needed if the cephalic vein is absent or if the vessel has a small caliber or a tortuous configuration. In order to preserve the entry vessel and resolve these difficulties, a standard algorithm has been proposed [16]. 

Four major improvements, which differ from previous studies, have been introduced. First, establish standard operation procedure and utilization priority of entry vessels. This is easier for inexperienced surgeons to follow and resolves difficulties when a target vessel cannot be easily identified. Second, use endovascular techniques to deal with target vessels of suboptimal quality, such as those of small-caliber or tortuous route [8,16]. This preserves patients’ native vessels and reduces difficulties during re-implantation. Echo-guided puncture at high neck areas, i.e., thyroid cartilage level, to reach the internal jugular vein (IJV) is only reserved for patients without accessible native vessels. Third, completely avoid subclavian vein and lower neck IJV puncture. This eliminates iatrogenic arterial puncture [13,14], hemothorax [17], and mediastinal hematoma [18]. In addition, pneumothorax [13,14] and catheter fracture related to pinch-off syndrome [19] are completely avoided. Fourth, enforce quality control by the addition of quantified catheter length formulae [20] for catheter length estimation and intra-operative fluoroscopy [21]. 

Based on these improvements, the standard algorithm for port implantation not only utilizes entry vessels more efficiently but also minimizes variations in implantation as a result of quantified implantation quality control. Our previous study showed good short-term results [16] and, in this study, we further analyzed long-term results of the standard algorithm for intravenous port implantation, not only to prove its reproducibility in various clinical scenarios but also to provide easy-to-follow recommendations for port implantation.

## 2. Materials and Methods

### 2.1. Patient Population

A total of 3144 patients who underwent intravenous port implantation between March 2012 and December 2018 at Chang Gung Memorial Hospital were enrolled. The patients’ disposition diagrams are shown in Appendix A. There were 2950 consecutive patients who underwent port implantation via the superior vena cava (SVC) route or the inferior vena cava (IVC) route (Appendix A). This study was approved by the Institutional Review Board under the approval numbers 20150929B0 and 201800329B0. Patients’ signed, informed consent was obtained before enrollment. This retrospective study was supported by Chang Gung Medical Foundation under grant number CMRPG5G0131.

### 2.2. Standard Algorithm for Port Implantation 

In order to better preserve the entry vessel and to improve convenience of nursing care, the SVC route was preferred (Figure 1). The IVC route was reserved for patients with SVC syndrome. Potential entry vessels for the SVC route were the cephalic vein, the deltoid branch of the thoracoacromial vein, and the IJV. Potential entry vessels for the IVC route were the greater saphenous vein (GSV) and the femoral vein. The pocket sites for port embedding of SVC and IVC ports were the fascia of the pectoralis major muscle and the abdominal rectus muscle near the anterior superior iliac spine (ASIS), respectively. Intra-operative fluoroscopy was used for catheter guidance and confirmation of tip location. Postoperative assessment of catheter-nut angle and catheter tip location on chest plain film was done in all patients.

### 2.3. Surgical Method

For SVC ports, the anatomic landmark was the coracoid process of the scapulae. After aseptic preparation, the coracoid process was identified and local anesthesia applied at this point. A 2-cm incision was made and deepened to the deltopectoral groove. Vessel cutdown was used for vessels with adequate caliber and smooth configuration. Wire assistance without puncture was used for vessels with adequate vessel caliber but with tortuous path. Wire assistance with puncture was used for vessels with both small-caliber and tortuous path. For patients in whom manual wire cannulation was difficult, the cannulation route was clarified on intra-operative venography. For patients without accessible cephalic or deltoid branch of the thoracoacromial vein, the IJV served as a rescue entry vessel, and echo-guided IJV high-neck puncture was used for implantation. For IVC ports, the anatomic landmarks were the anterior inferior iliac spine and the pubic tubercle, and two small incisions were made at the anterior inferior iliac spine and the subinguinal area. Greater saphenous vein exploration and catheter implantation were done via the subinguinal incision, and the port was fixed over the abdominal rectus fascia via incision at the ASIS. 

### 2.4. Follow-Up and Postoperative Surveillance

All patients underwent plain chest radiography following the procedure in order to confirm the catheter tip location, catheter-nut angle, and integrity of the implanted port. Catheter-nut angle and tip location measurements were recorded in a picture archiving and communication system (PACS). All patients returned to the outpatient department for follow-up at 3-month intervals and underwent flushing for maintenance. Functional period was defined as the time during which the implanted port maintained normal function. For patients without complications, this was from implantation date to date of removal or last follow-up. For those with complications needing re-intervention, it was from implantation date to date of re-intervention.

### 2.5. Statistics

All collected data were first analyzed using univariate analysis. Categorical variables were compared using chi-square test or Fisher’s exact test. A *p*-value of less than 0.05 was considered statistically significant. Confidence intervals (CI) were assumed to have a coverage probability of 95%. Complication rates are presented as episode percentage among the whole population and incidence is presented as episode per 1000 catheter days. Loess smooth and linear regression curve models were deployed for analysis of operation time variation. All analyses were performed using SAS, version 9 (SAS Institute, Cary, NC, USA).

## 3. Results

### 3.1. Sex Differences 

The mean age of female patients was younger than that of male patients (57.3 ± 12.8 years vs. 59.6 ± 12.8 years, *p* < 0.0001) (Table 1). More male patients had head and neck cancer (*p* < 0.0001), and more female patients had thoracic malignancy (*p* < 0.0001). In addition, male patients had greater mean body height (165.8 ± 7.2 cm vs. 154.9 ± 6.9 cm, *p* < 0.001) and mean body weight (64.4 ± 11.8 kg vs. 55.8 ± 10.4 kg, *p* < 0.001) compared to female patients, though mean BMI was similar for the two genders (*p* = 0.2941). The left-side approach was used more in female patients. Operation methods differed significantly between male and female patients (*p* < 0.0001). Mean catheter-nut angle of male and female patients was 169.7 ± 7.7° and 170.0 ± 7.2°, respectively (*p* = 0.1926). Deeper mean catheter tip location was noted in female patients (1.4 ± 1.5 cm, *p* < 0.0001). Mean functional period of ports in male and female patients was 458.8 ± 449.3 and 658.7 ± 535.7 days, respectively (*p* < 0.0001).

### 3.2. Procedure and Late Complications

Overall complication rate was 3.08% with an incidence of 0.057 episodes per 1000 catheter days. Only two patients were identified with port rotation, which resulted from the cutting through of thin pectoralis major muscle by the stay suture during postural movement. No pneumothorax, hematoma, catheter kinking, catheter fracture, or pocket erosion was identified. Procedure-related complication rate was 0.07% (Table 2). The complication rates of migration, malfunction, infection, and deep vein thrombosis (DVT) were 0.68%, 0.54%, 1.29%, and 0.51%, respectively. The incidences of port rotation, migration, malfunction, infection, and DVT were 0.001, 0.012, 0.010, 0.024, and 0.009 per 1000 catheter days, respectively (Table 2). Compared with our previous study, procedure-related complications were nearly completely eliminated except for port rotation Appendix A. Late complication rate and incidence were markedly lower compared with our previous study (Figure 2A,B). Relative risk ratios of migration, malfunction, infection, and DVT were 0.288, 0.148, 0.137, and 0.432, respectively (Figure 2C).

### 3.3. Evolution of Operation Method and Operative Time 

From 2012 to 2018, an increasing number of procedures was done by vessel cutdown (*p* = 0.0146) (Table 3). The percentage of procedures using vessel cutdown increased from approximately 60% in earlier years to 74.3% in 2018. In addition, echo-guided IJV high-neck puncture markedly declined from 4.7% to 1.0% (*p* = 0.027). The highest percentage (27.2%) of wire assistance without puncture was reported in 2013, and gradually decreased thereafter. The variation trend in operative time was further analyzed using Loess smooth curve and linear regression models and showed gradual decline from 37.88 min to 23.20 min. In the linear regression curve model, the operation time also declined from 34.33 min to 22 min (Figure 2D).

## 4. Discussion 

In the study cohort, disease distribution was similar in both sexes except for thoracic malignancy and head and neck cancers. More female patients had thoracic malignancy, since we categorized breast cancer as a thoracic malignancy (*p* < 0.0001), whereas more male patients had head and neck cancers. These differences may have led to more left-side implantations (*p* < 0.0001) and slightly deeper catheter tip locations (*p* < 0.001) in females compared to males. The variations in operation method may be related to the variations in individual entry vessels, such as three-dimensional spatial orientation of blood vessels and vessel caliber. More high-neck echo-guided IJV puncture was reported in female patients because of inadequate vessel quality.

In this study, we used the coracoid process of the scapula as a landmark for incision, thus minimizing incision variation [20,22]. In addition, vessel cutdown was the preferred method and the cephalic vein was the target vessel for port implantation because of its low complication rate [15,23]. In patients with inaccessible cephalic vein, the thoracoacromial vein may serve as an alternative choice [8,24]. In addition, anatomic variations such as small vessel caliber and tortuous path may prohibit direct catheter implantation. To overcome these difficulties and to minimize IJV puncture, endovascular techniques (Appendix A) may be employed for assistance. With the aid of these endovascular techniques, port implantation can be done even when the quality of the entry vessel is suboptimal. Echo-guided high-neck IJV puncture was reserved only for patients with no accessible vessels. These differences not only eliminated pneumothorax, arterial puncture, and catheter fracture but also reduced IJV puncture from 4.7% to less than 1.0% (*p* = 0.027). With the aid of the standard algorithm, average operative time gradually decreased in each subsequent year for all methods (*p* < 0.0001), and this trend was confirmed using Loess smooth and linear regression curve models. 

The pocket was created of adequate size along the avascular plane between the pectoralis major muscle fascia and subcutaneous tissue. This strategy not only preserved the whole layer of subcutaneous tissue but also avoided catheter impingement. By adhering to the algorithm, hematoma, port erosion, impingement-related fracture [25], and malfunction were markedly decreased [6]. Intra-operative fluoroscopy not only allowed real-time monitoring of port implantation, it also helped to standardize the catheter-nut angle and to optimize the tip location [5,6]. Migration was minimized by optimal tip location; however, it still occurred because the intravascular portion of the catheter was free within the vessel to move upward with increased intrathoracic pressure [20,26]. Quality control was achieved by standardized operative procedures and planning, the use of intra-operative fluoroscopy monitoring, and post-operative chest plain film confirmation. Low complication rate and incidence and stable implantation quality were re-confirmed, revealing the reproducibility of the standard algorithm.

There are some limitations to this study. This study was a retrospective study; however, the large number of cases who received intravenous port implantation by the standard algorithm allowed for homogeneity in the management of the whole study cohort and reliable results. Second, a functional intravenous port not only relies on good implantation quality but also depends on good maintenance strategy, and this may explain why infection and malfunction remained, even after the standard algorithm was implemented in practice. Appropriate irrigation volume and maintenance strategy may be needed to reduce the occurrence [27,28] and further investigation is warranted. Third, posture variations resulting in changes of vascular caliber may confound the analysis of DVT occurrence. From the literature review, the occurrence of DVT has been associated with catheter–vessel caliber ratio, i.e., cross-sectional area of catheter/cross-sectional area of vessel [29]. Lengthening of the subclavian vein and, thus, a decrease in catheter–vessel caliber ratio can result during stretch posture and result in DVT. Further investigations are warranted. Despite these limitations, the study not only shows that the standard algorithm for port implantation works to minimize complications, but it also confirms its reproducibility.

## 5. Conclusions

The proposed standard algorithm and the described surgical planning and techniques may help surgeons minimize the occurrence of complications related to intravenous port implantation.

## Figures and Tables

**Figure 1 jpm-11-00344-f001:**
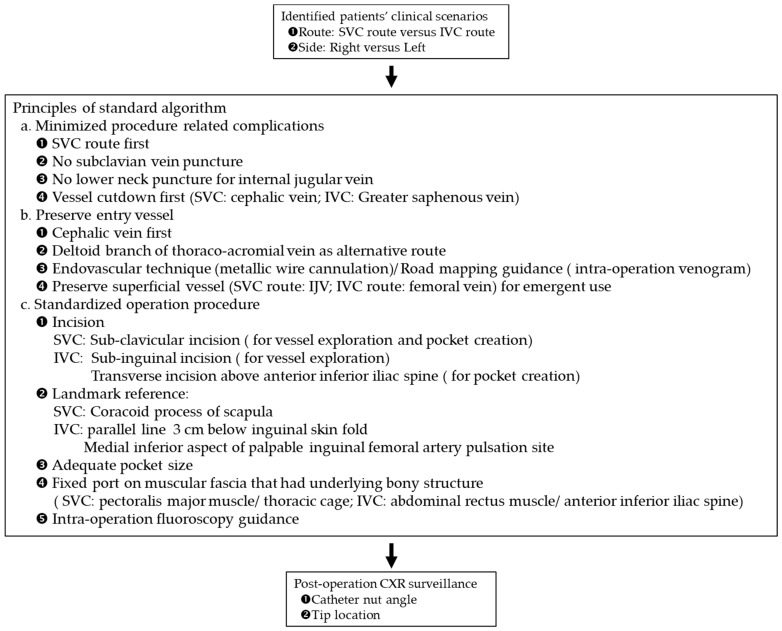
Standard algorithm of intravenous port implantation.

**Figure 2 jpm-11-00344-f002:**
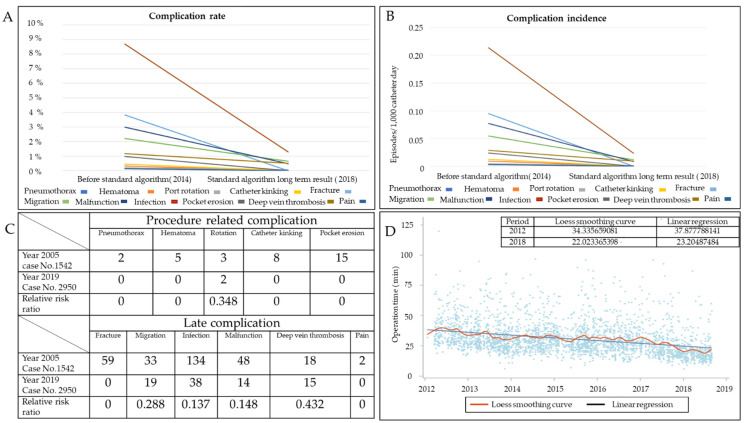
Variation trends of complication rate, incidence, and operation. (**A**) Variation trends of procedure- and catheter-related complication rates. (**B**) Variation trend of procedure- and catheter-related complication incidences. (**C**) Relative risk ratio between Year 2005 and Year 2018 cohorts. (**D**) Variation trend of operation time.

**Table 1 jpm-11-00344-t001:** Patient characteristics (male versus female).

Variables	*n* (%) or Mean ± SD		*p* Value
	Male	Female	
Age	59.6 ± 12.8	57.3 ± 12.8	<0.0001
Body height	165.8 ± 7.2	154.9 ± 6.9	<0.0001
Body weight	64.4 ± 11.8	55.8 ± 10.4	<0.0001
Body mass index	23.4 ± 3.8	23.2 ± 4.0	0.2941
Malignancy ^1^			
Head and neck	264 (15.4%)	38 (3.07%)	<0.0001
Thorax	653 (38.1%)	621 (50.20%)	<0.0001
Abdomen	662 (38.6%)	471 (38.08%)	0.7536
Pelvis	7 (0.4%)	6 (0.49%)	0.7572
Soft tissue	9 (0.5%)	8 (0.65%)	0.6675
Hematology	146 (8.5%)	104 (8.41%)	0.9114
Other	15 (0.9%)	6 (0.49%)	0.2130
Side			
Right	1618 (94.4%)	1022 (82.6%)	<0.0001
Left	95 (5.6%)	215 (17.4%)	
Entry route			
Superior vena cava	1683 (98.2%)	1222 (98.8%)	0.2388
Inferior vena cava	30 (1.8%)	15 (1.2%)	
Entry vessel			
Cephalic vein	1467 (85.7%)	1034 (83.6%)	
Thoracoacromial vein	177 (10.3%)	149 (12.0%)	0.0632
IJV ^2^	36 (2.1%)	39 (3.2%)	
Other ^3^	33 (1.9%)	15 (1.2%)	
Port type			0.0100
B’Braun Fr. 6.5	528 (30.8%)	331 (26.7%)	
Bard X Fr.6/8	408 (23.8%)	357 (28.9%)	
Bard power Fr.6	544 (31.8%)	388 (31.4%)	
Polysite Fr.7	233 (13.6%)	161 (13.0%)	
Operation method			<0.0001
Vessel cutdown	1085 (63.4%)	785 (63.5%)	
Wire assistance without puncture	394 (23.0%)	208 (16.8%)	
Wire assistance with puncture	173 (10.1%)	180 (14.5%)	
Wire and venogram assistance			
a. Without puncture	7 (0.4%)	6 (0.5%)	
b. Puncture	19 (1.1%)	18 (1.5%)	
Echo guide puncture	35 (2.0%)	40 (3.2%)	
Operation time (Minutes)			
Vessel cutdown	26.4 ± 10.7	28.6 ± 10.8	<0.0001
Wire assistance without puncture	29.9 ± 10.7	30.5 ± 10.3	0.4947
Wire assistance with puncture	39.2 ± 14.9	41.5 ± 16.6	0.1748
Wire and venogram assistance			
a. Without puncture	32.4 ± 11.1	34.2 ± 8.9	0.7635
b. Puncture	48.9 ± 17.0	48.3 ± 16.4	0.9038
Echo guide puncture	62.0 ± 12.6	60.2 ± 22.2	0.6585
Post-op quality			
Catheter-nut angle (°)	169.7 ± 7.7	170.0 ± 7.2	0.1926
Tip location (cm)	1.0 ± 1.4	1.4 ± 1.5	<0.0001
Functional period (days)	458.8 ± 449.3	658.7 ± 535.7	<0.0001
Follow-up status			
Alive	949 (55.4%)	864 (69.9%)	<0.0001
Expired	514 (30.0%)	254 (20.5%)	
Discharged Against advice	250 (14.6%)	119 (9.6%)	

^1^ Forty-three male patients have diagnosed double cancers, 17 female patients have diagnosed double cancers. ^2^ IJV: internal jugular vein. ^3^ Other: 38 right greater saphenous veins, 7 left greater saphenous veins, 1 right axillary vein, 2 right external jugular veins.

**Table 2 jpm-11-00344-t002:** Complication rate, incidence, and reason for re-intervention.

**Entry Vessel**	**Cephalic Vein**		**Thoracoacromial Vein**		**Internal Jugular Vein**		**Other**		**Greater Saphenous Vein**		**Total**
Number of patients	2242	259	296	30	61	14	48	0	38	7	2950
Side of Complication	Right	Left	Right	Left	Right	Left	Right	Left	Right	Left	91
Procedure related											
Rotation	1	0	0	0	1	0	0	0	0	0	2
Late											
Infection	29	3	4	0	1	0	0	0	1	0	38
Malfunction	7	1	2	0	2	0	2	0	1	1	16
Migration	16	0	3	0	0	0	1	0	0	0	20
Deep vein thrombosis	10	4	1	0	0	0	0	0	0	0	15
**Complication Rate**											
**Entry Vessel**	**Cephalic Vein**		**Thoracoacromial Vein**		**Internal Jugular Vein**		**Other**		**Greater Saphenous Vein**		**Total**
Number of patients	2242	259	296	30	61	14	3	0	38	7	2950
Side/Total rate	Right	Left	Right	Left	Right	Left	Right	Left	Right	Left	3.08%
Procedure related											
Rotation	0.04%	0	0	0	1.64%	0	0	0	0	0	0.07%
Late											
Infection	1.29%	1.16%	1.35%	0	1.64%	0	0	0	2.63%	0	1.29%
Malfunction	0.31%	0.39%	0.68%	0	3.28%	0	4.16%	0	2.63%	14.29%	0.54%
Migration	0.71%	0	1.01%	0	0	0	2.08%	0	0	0	0.68%
Deep vein thrombosis	0.45%	1.54%	0.34%	0	0	0	0	0	0	0	0.51%
**Incidence**											
**Entry Vessel**	**Cephalic Vein**		**Thoracoacromial Vein**		**Internal Jugular Vein**		**Other**		**Greater Saphenous Vein**		**Total**
Sum of catheter days	1,188,218	180,844	145,804	23,159	38,196	8788	4945	0	8612	2142	1,600,708
Side/Total incidence	Right	Left	Right	Left	Right	Left	Right	Left	Right	Left	0.057
Procedure related											
Rotation	0.001	0	0	0	0.026	0	0	0	0	0	0.001
Late											
Infection	0.024	0.017	0.027	0	0.026	0	0	0	0.116	0	0.024
Malfunction	0.006	0.006	0.014	0	0.052	0	0.404	0	0.116	0.467	0.010
Migration	0.013	0	0.021	0	0	0	0.202	0	0	0	0.012
Deep vein thrombosis	0.008	0.022	0.007	0	0	0	0	0	0	0	0.009

Incidence rate = complication case/sum of person-days × 1000. Other: external jugular vein/axillary vein.

**Table 3 jpm-11-00344-t003:** Evolution of operation method and operative time.

	**Year**	**2012**	**2013**	**2014**	**2015**	**2016**	**2017**	**2018**	***p* Value ^a^**
**Case Number**									
Vessel cutdown		211 (65.7%)	313 (59.5%)	304 (61.3%)	277 (60.8%)	283 (66.4%)	257 (60.9%)	225 (74.3%)	0.0146
Wire assistance without puncture		57 (17.8%)	143 (27.2%)	113 (22.8%)	90 (19.7%)	73 (17.1%)	80 (19.0%)	46 (15.2%)	0.0014
Wire assistance with puncture		38 (11.8%)	52 (9.9%)	64 (12.9%)	78 (17.1%)	50 (11.7%)	52 (12.3%)	19 (6.3%)	0.3424
Wire and venogram assistance									
a. Without puncture		0 (0.00%)	0 (0.00%)	1 (0.2%)	0 (0.0%)	3 (0.7%)	8 (1.9%)	1 (0.3%)	-
b. Puncture		0 (0.00%)	0 (0.00%)	1 (0.2%)	3 (0.7%)	9 (2.1%)	15 (3.5%)	9 (3.0%)	-
Echo guide puncture		15 (4.7%)	18 (3.4%)	13 (2.6%)	8 (1.7%)	8 (1.9%)	10 (2.4%)	3 (1.0%)	0.0027
	**Year**	**2012**	**2013**	**2014**	**2015**	**2016**	**2017**	**2018**	***p* Value ^b^**
**Operation Time**									
Vessel cutdown		34.2 ± 10.5	28.9 ± 8.8	29.7 ± 10.2	28.5 ± 11.7	27.8 ± 10.3	22.8 ± 9.6	18.5 ± 7.5	<0.0001
Wire assistance without puncture		38.8 ± 9.5	33.0 ± 9.3	31.7 ± 10.2	29.9 ± 11.2	29.0 ± 7.6	23.5 ± 9.4	20.4 ± 6.1	<0.0001
Wire assistance with puncture		48.1 ± 11.2	45.1 ± 12.9	41.5 ± 15.4	38.8 ± 15.4	40.6 ± 16.2	36.1 ± 19.1	25.2 ± 7.0	<0.0001
Wire and venogram assistance									
a. Without puncture		-	-	43.0 *	-	27.7 ± 11.0	33.9 ± 10.0	35.0 *	-
b. Puncture		-	-	58.0 *	70.7 ± 21.8	50.8 ± 15.1	45.9 ± 15.2	42.7 ± 14.9	-
Echo guide puncture		59.3 ± 23.0	61.0 ± 21.4	60.1 ± 7.6	67.7 ± 16.2	67.1 ± 22.6	53.3 ± 11.3	64.7 ± 20.1	0.9509

*: one patient, mean operation time. -: no data available. ^a^: analyzed by logistic regression (trend test). ^b^: analyzed by linear regression (trend test).

## Data Availability

The data presented in this study are available on request from the corresponding author. The data are not publicly available due to limitation of Personal Data Protection Act of Taiwan.

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
