# Peer review of "Long-Term Results of a Standard Algorithm for Intravenous Port Implantation"

_jpm, 2021, doi:10.3390/jpm11050344_

Round 1

Reviewer 1 Report

There are some not defined abbreviations, like IJV.
Line 220 looks incomplete.
It will be important to include the algorithm as a Figure in the text, not as supplementary information.

Author Response

Reviewer 1

Comments

Reply

1.There are some not defined abbreviations, like IJV.

Thanks for your comment. We’ve modified and added a summary of abbreviations.

2.Line 220 looks incomplete.

Thanks for your comment. We’ve modified the sentence.

3.It will be important to include the algorithm as a Figure in the text, not as supplementary information.

Thanks for your comment. We have modified following the reviewer’s comment.

Reviewer 2 Report

Congratulations to authors on conducting this important study with a relatively large sample size. Very impressive low procedure-related complication rate. Improved operative time from 2012 to 2018.

The command of English throughout the manuscript is very poor and makes the manuscript difficult to understand. The manuscript would significantly benefit from proofreading and grammatical corrections from a native English language speaker.

Several suggestions:

Line 22: Most studies have focused on different approach without using a systematic -> Most studies have not used a systematic

Line 23: long term result -> long-term results

Line 26: functional outcomes - what does this mean?

Lines 27-28: wire assistance without puncture - what does this mean?

Lines 30-31: Very surprising to have zero pneumothorax from 2950 patients?

Line 46: direct -> directly

Lines 59-60: Completed avoidance risk factors that correlated complication -> unclear what this means

Line 60: prohibition subclavian vein -> unclear what this means

Line 63: add -> addition

Line 65: intra-operation -> intra-operative

Line 71: 2950 patients - were these ALL consecutive patients? 

Lines 125-126: add units 'years'

Line 132: significant -> significantly

Line 132: catheter-nut -> what does this mean?

Line 135: functional period -> what does this mean?

Table 1: Add space between N(%). Add abbreviations list for IJV. For (%) one decimal place sufficient. For operation time, one decimal place sufficient. Difficult to read when Table 1 is discontinuous across two pages - try to display within a single page if possible.

Table 2: reason of reintervention -> reason for reintervention. For % one decimal place sufficient. Add abbreviations list for IJV, EJV, GSV.

Figure 1: Hard to read words/numbers too small.

Supplement 1. Title says 'Inclusion and exclusion criteria' however it looks more like a CONSORT diagram / patient disposition diagram rather than explicitly stating a list of inclusion and exclusion criteria.

Supplement 3. Some numbers highlighted grey whilst others not - ensure consistent. For % one decimal place sufficient. For operation time numbers one decimal place sufficient. 

Supplement 6. B. White arrow not seen?

Author Response

Reviewer 2

Comments

Reply

The manuscript would significantly benefit from proofreading and grammatical corrections from a native English language speaker.

Thanks for your comment. We have sent the document for certified English editing.

Line 22: Most studies have focused on different approach without using a systematic -> Most studies have not used a systematic

Thanks for your comment. We have re-written in accordance with the reviewer’s comment.

Line 23: long term result -> long-term results

Thanks for your comment. We have corrected this.

Line 26: functional outcomes - what does this mean?

Thanks for your comment. We have modified the “functional outcomes” to “long-term outcomes”.

Lines 27-28: wire assistance without puncture - what does this mean?

Thanks for your comment. We have included a clearer description as follows:

“In female patients, wire assistance without use of puncture sheath was less likely and echo-guided puncture via internal jugular vein (IJV) with use of puncture sheath was more likely to be performed, compared to male patients.”

Lines 30-31: Very surprising to have zero pneumothorax from 2950 patients?

Thanks for your comments. According to the literature review, pneumothorax and hemothorax are iatrogenic complications related to subclavian puncture for the subclavian vein, and lower neck puncture for the internal jugular vein [1-4]. In our series, the majority of patients received port implantation via a native vessel but we did not perform subclavian vein puncture. For those in whom manual implantation was difficult, wire assisted endovascular techniques with puncture sheath were used to establish an entry route. With the aid of this technique, the number of patients who needed echo guided puncture was further lowered to 0.99%. In addition, internal jugular vein puncture at thyroid cartilage level. ie. high neck entry site, was recommended only for patients who lacked a suitable native vessel for intravenous port implantation. With the aid of IJV high neck echo guided puncture, the tip of the puncture needle entered the vessel at the lower neck portion of the IJV that is still outside the thoracic inlet.  Thus the possibility of pneumothorax and hemothorax was completely avoided, resulting in no pneumothorax or hemothorax.

Reference

1.        McGee DC, Gould MK. Preventing complications of central venous catheterization. N Engl J Med 2003;348:1123-33

2.        Kusminsky RE. Complications of central venous catheterization. J Am Coll Surg 2007;204:681-96

3.        Troianos CA, Hartman GS, Glas KE, et al. Guidelines for performing ultrasound guided vascular cannulation: recommendations of the American Society of Echocardiography and the Society of Cardiovascular Anesthesiologists. J Am Soc Echocardiogr 2011;24:1291-318

4.        Orsi F, Grasso RF, Arnaldi P, et al. Ultrasound guided versus direct vein puncture in central venous port placement. J Vasc Access 2000;1:73-7.

Line 46: direct -> directly

Thanks for your comment.  We have corrected this.

Lines 59-60: Completed avoidance risk factors that correlated complication -> unclear what this means

Thanks for your comment. We have rewritten the introduction to clarify.

Line 60: prohibition subclavian vein -> unclear what this means

Thanks for your comment. We have rewritten to clarify.

Line 63: add -> addition

Thanks for your comment. We have corrected this.

Line 65: intra-operation -> intra-operative

Thanks for your comment. We have corrected this.

Line 71: 2950 patients - were these ALL consecutive patients? 

Thanks for your comment. All 2950 patients were consecutive and underwent port implantation under the same algorithm.

Lines 125-126: add units 'years'

Thanks for your comment.  We have modified following the reviewer’s comment.

Line 132: significant -> significantly

Thanks for your comment. We have corrected this.

Line 132: catheter-nut -> what does this mean?

Thanks for your comment. Catheter-nut angle mean the actual curvature between locking nut and proximal end of catheter. The angle was the angle between line A and B.  Line A (Blue line) is the central line of locking nut. Line B (Black line) was the draw line between point  and ‚. The point  is the center line of the locking nut at the exit point of the catheter.  The point ‚ is the point of tangency of the curvature site of proximal end of catheter.(Red line)    Connecting these 2 points and line B (Black line) is drawn.   The angle should be an obtuse angle that not only provide flexibility for shoulder movement but also keep catheter lumen without compression.  This would result in well maintenance result and keep well catheter function.[1]

Reference

1.Wu CY, Hu HC, Ko PJ, Fu JY, Wu CF, Liu YH, Li HJ, Kao TC, Yu SY, Chang CJ, Hsieh HC. Risk factors and possible mechanisms of superior vena cava intravenous port malfunction Ann Surg. 2012 May;255(5):971-5

Line 135: functional period -> what does this mean?

Thanks for your comment. We defined functional period as the time the implanted port maintained normal function. For patients without complications, it was the time from implantation date to date of removal or last follow up date. For those with complications needing re-intervention, it was the time from implantation date to date of re-intervention. We have added the definition of functional period in 2.4 follow up and post-operation surveillance.

Table 1: Add space between N(%). Add abbreviations list for IJV. For (%) one decimal place sufficient. For operation time, one decimal place sufficient. Difficult to read when Table 1 is discontinuous across two pages - try to display within a single page if possible.

Thanks for your comments. We have modified following the reviewer’s comment. The revised table is included in the revised manuscript.

Table 2: reason of reintervention -> reason for reintervention. For % one decimal place sufficient. Add abbreviations list for IJV, EJV, GSV.

Thanks for your comments. We have modified the table in accordance with the reviewer’s comment.  The revised table is included in the revised manuscript.

Figure 1: Hard to read words/numbers too small.

Supplement 1. Title says 'Inclusion and exclusion criteria' however it looks more like a CONSORT diagram / patient disposition diagram rather than explicitly stating a list of inclusion and exclusion criteria.

Thanks for your comments. The words/ numbers have been modified, and the revised figure is as follows. In addition, the title of the supplement has also been revised to reflect the reviewer’s comment.

Supplement 3. Some numbers highlighted grey whilst others not - ensure consistent. For % one decimal place sufficient. For operation time numbers one decimal place sufficient. 

Thanks for your comment. We have edited the document following the reviewer’s comment.

Supplement 6. B. White arrow not seen?

Thanks for your comment. We have corrected this.

Reviewer 3 Report

The authors have studied the long term results of port implantation following a standard algorithm. They studied 2950 patients who had gone through intravenous port implantation during the years 2012 through 2018. Safety and functional outcomes were studied. They claim that the proposed standard algorithm and the surgical planning may help surgeons to reduce complications after intravenous port implantation. 

Though this study is retrospective, it is a useful tool to understand how the intravenous implantations could lead to infection and malfunction and how we can reduce it using the standard algorithm.

I have minor revision as the grammar and english could be improved. 

For major revision: Introduction must be rewritten explaining in more details of why this study is necessary and how is it useful for the future.

Author Response

Reviewer 3

Comments

Reply

I have minor revision as the grammar and english could be improved.

Thanks for your comment. We have sent the document for certified English editing.

Introduction must be rewritten explaining in more details of why this study is necessary and how is it useful for the future.

Thanks for your comment. We have rewritten the introduction to reflect the reviewer’s comment.

Round 2

Reviewer 2 Report

The article is much improved following revisions. Congratulations to the authors on presenting interesting findings from their important study.

Suggest reduce decimal places, where possible, throughout text and in Tables 1 & 3 to make easier to read. Please make font sizes consistent in Table 1 (some fonts appear larger than others). Suggest enlarging Figure 2 panels which appear quite small. Suggest remove decimal places for Figure 2 panel A (e.g. 10% instead of 10.00%). 

Author Response

Reviewer 1

Comments

Reply

The article is much improved following revisions. Congratulations to the authors on presenting interesting findings from their important study.

Thanks for your comments

Suggest reduce decimal places, where possible, throughout text and in Tables 1 & 3 to make easier to read. Please make font sizes consistent in Table 1 (some fonts appear larger than others).

Thanks for your comments

We’ve modified as reviewers’ comments

Suggest enlarging Figure 2 panels which appear quite small. Suggest remove decimal places for Figure 2 panel A (e.g. 10% instead of 10.00%).

Thanks for your comments

We’ve modified as reviewers’ comments

Reviewer 3 Report

The authors have addressed the major revision by writing and explaining more in the introduction part.

Goodluck.

Author Response

Reviewer 2

The authors have addressed the major revision by writing and explaining more in the introduction part.

Goodluck.

Thanks for your comments